# AI-Based CXR First Reading: Current Limitations to Ensure Practical Value

**DOI:** 10.3390/diagnostics13081430

**Published:** 2023-04-16

**Authors:** Yuriy Vasilev, Anton Vladzymyrskyy, Olga Omelyanskaya, Ivan Blokhin, Yury Kirpichev, Kirill Arzamasov

**Affiliations:** 1State Budget-Funded Health Care Institution of the City of Moscow “Research and Practical Clinical Center for Diagnostics and Telemedicine Technologies of the Moscow Health Care Department”, Petrovka Street, 24, Building 1, 127051 Moscow, Russia; 2Department of Information and Internet Technologies, I.M. Sechenov First Moscow State Medical University of the Ministry of Health of the Russian Federation (Sechenov University), Trubetskaya Street, 8, Building 2, 119991 Moscow, Russia

**Keywords:** AI for chest X-ray first reading, external validation, local test set, prospective validation

## Abstract

We performed a multicenter external evaluation of the practical and clinical efficacy of a commercial AI algorithm for chest X-ray (CXR) analysis (Lunit INSIGHT CXR). A retrospective evaluation was performed with a multi-reader study. For a prospective evaluation, the AI model was run on CXR studies; the results were compared to the reports of 226 radiologists. In the multi-reader study, the area under the curve (AUC), sensitivity, and specificity of the AI were 0.94 (CI95%: 0.87–1.0), 0.9 (CI95%: 0.79–1.0), and 0.89 (CI95%: 0.79–0.98); the AUC, sensitivity, and specificity of the radiologists were 0.97 (CI95%: 0.94–1.0), 0.9 (CI95%: 0.79–1.0), and 0.95 (CI95%: 0.89–1.0). In most regions of the ROC curve, the AI performed a little worse or at the same level as an average human reader. The McNemar test showed no statistically significant differences between AI and radiologists. In the prospective study with 4752 cases, the AUC, sensitivity, and specificity of the AI were 0.84 (CI95%: 0.82–0.86), 0.77 (CI95%: 0.73–0.80), and 0.81 (CI95%: 0.80–0.82). Lower accuracy values obtained during the prospective validation were mainly associated with false-positive findings considered by experts to be clinically insignificant and the false-negative omission of human-reported “opacity”, “nodule”, and calcification. In a large-scale prospective validation of the commercial AI algorithm in clinical practice, lower sensitivity and specificity values were obtained compared to the prior retrospective evaluation of the data of the same population.

## 1. Introduction

Respiratory diseases are among the leading causes of death [1]. Chest radiography (CXR) is the most common method of medical imaging. With a high degree of standardization and apparent simplicity of CXR, sensitivity is of great importance. The “double reading” method is used to minimize the false-negative results [2,3]. 

During the SARS-CoV-2 (COVID-19) pandemic, the burden on the healthcare system was manifested by a shortage of medical personnel, with the increased number of imaging studies, leading to a reduction in the time available for reporting, which pushes the increase in human errors [4].

In this situation, algorithms based on artificial intelligence (AI) capable of detecting pathology in CXR are of great practical importance. CXR meets the three main criteria for deploying AI systems: mass use, relative standardization, and digitalization. Furthermore, an AI system can serve as a clinical decision support system (CDSS) [5,6]. 

The diagnostic accuracy of the algorithms provided by the developers is quite high [7,8,9], reaching the same accuracy for radiologists [10], and for some solutions even exceeding them [11,12]. As of the beginning of 2023, 29 AI-based software products have European certification for medical use as a medical device (CE MDR/MDD), of which 11 have passed a similar certification in the United States [13]. It is important to note that one such software product, approved for circulation as a medical device, is intended for a completely autonomous analysis of CXR [14]. This AI algorithm sorts examinations, detects CXR without pathology, and forms a complete description protocol that does not require validation by a radiologist; this approach reduces the burden on the radiologist, allowing them to focus on cases with pathologies [11].

The problem is that most of the work on assessing the diagnostic accuracy of AI algorithms for CXR indicates metrics obtained by developers on limited datasets in the so-called “laboratory conditions”. As can be seen from recent studies [12,15], the metrics obtained in this way look attractive for the subsequent implementation of such algorithms in clinical practice. Will AI for CXR analysis also work well and demonstrate high diagnostic accuracy metrics in real clinical practice? There are very few large-scale works on this topic; it is possible to point out only a few studies [11,16]. Using recent research as an example, we can see the introduction of autonomous AI into clinical practice, but the issue of threshold values for diagnostic accuracy metrics remains open.

Despite the extensive possibilities of using AI for CXR analysis [17], the influence of surrounding clinical information on the result remains understudied, especially given the trend toward controlling systematic errors in radiology [18].

We assessed the practical and clinical efficacy of the AI algorithm for CXR analysis on external data by conducting a two-stage multicenter study: a retrospective case-control study and a prospective validation study.

## 2. Materials and Methods

This study includes the data from a registered clinical trial NCT04489992. The local independent Ethics Committee approved the study, and all data were de-identified. The overall study flowchart is presented in Figure 1.

The study consisted of two parts: a retrospective evaluation (right side of Figure 1) and a prospective evaluation (left side of Figure 1) of the diagnostic accuracy metrics of the AI system. The retrospective study was designed as an international multi-reader study, which has been performed to determine an average diagnostic accuracy of radiologists interpreting chest X-ray images and AI system performance metrics for the same use cases. In a prospective study, patient studies were processed by an on-stream AI system. The result of the work of the AI system was compared for the entire volume of studies with the conclusion of a radiologist. To ensure quality control, a sample of prospective studies was formed and was sent for expert evaluation of the correctness of the initial description of the study by a radiologist. The obtained values of diagnostic accuracy metrics were compared with those for a retrospective study.

### 2.1. AI System

A certified, commercially available AI system was used: Lunit INSIGHT CXR for Chest Radiography Version 3.110, Lunit, Seoul, Republic of Korea. The AI system used a ResNet34-based deep convolutional neural network with 10 different abnormality specific channels in the final layer [19] Additional information about the AI system is available on the official website [20]. The AI system detects ten radiological findings: atelectasis, calcification, cardiomegaly, consolidation, fibrosis, mediastinal widening, nodule, pneumothorax, pleural effusion, and pneumoperitoneum.

### 2.2. Retrospective Evaluation

A retrospective evaluation was performed by a multi-reader study to compare the diagnostic performance of AI and radiologists (see Table 1, column ‘Local test set’). A total of 160 certified radiologists participated in the study {years of experience [0–1): 39, [1–5): 49, [5–10): 30, 10+: 42} and interpreted 73 cases from a retrospectively collected local test set containing imaging data of the Moscow population. Each case was evaluated 46 times on average. The scoring was conducted using a 4-point scale: (1) definitely without pathology (probability of pathology—0%); (2) probably without pathology (probability of pathology—33%); (3) probably with pathology (probability of pathology—66%); (4) definitely with pathology (probability of pathology—100%). A consensus score per case was set based on the median reader score. In the case of a tie of frequencies, the higher score was selected. The readers were supplied with the patient sex and age information while reading the studies.

The local test set has been collected retrospectively from the radiological studies performed in outpatient Moscow state medical facilities for screening and diagnostic purposes in 2018–2019. These studies were obtained from 38 diagnostic devices from 7 vendors (Table 1). The dataset contained studies marked “without target pathology” and “with target pathology.” Target pathology is defined as a list of radiological findings based on clinical significance and frequency of occurrence in the routine practice of multidisciplinary medical facilities. All studies were pre-selected randomly based on electronic medical records and then evaluated in double consensus by radiology experts with at least five years of experience in thoracic radiology.

Inclusion criteria for the studies: (1) age over 18 years old; (2) DICOM format and anonymized; (3) an anterior-posterior/posterior-anterior view; (4) available results of AI analysis, (5) double consensus between two expert radiologists on the presence or absence of the target pathology.

Exclusion criteria: (1) post-lung surgery, including one lung remaining; (2) additional opacifications from medical devices, clothes, and extracorporeal objects; (3) technical defects or incorrect positioning; (4) absence of the expert consensus.

The list of the pathological CXR findings was used with terminology proposed by the Fleischner society [21] and completed with common significant findings out of the glossary:Pneumothorax;Atelectasis;Parenchymal opacification;Infiltrate or consolidation (infiltrate remains a controversial but applied term);Miliary pattern, or dissemination;Cavity;Pulmonary calcification;Pleural effusion;Fracture, or cortical irregularity.

More details on the local test set are presented in Table 1.

### 2.3. Prospective Evaluation

For a prospective evaluation, the AI was deployed for “research purposes only” with DICOM secondary capture (SC) available for each processed study in 10 inpatients and 97 outpatient departments in Moscow (Russia) via Unified Radiological Information Service (URIS), connecting all the state radiology sites of the Moscow Healthcare department [22,23]. The AI model was used to analyze incoming CXR studies during November and December 2020, and its results were compared to the reports of 226 radiologists. Only DICOM files were provided to AI. The AI consecutively analyzed 5373 studies. Studies were automatically routed to the AI using the following algorithm: (1) the study performed on the diagnostic device was sent to URIS and simultaneously became available for downloading and processing by the AI; (2) after processing the study, the AI returned the DICOM SC to URIS. Studies were processed sequentially from the queue. Inclusion criteria for the studies: (1) age over 18 years old; (2) DICOM format and anonymized; (3) an anterior-posterior/posterior-anterior view; (4) available results of AI analysis. A total of 621 were later excluded from the prospective evaluation (Figure 1) due to the technical defects, namely a lateral view or patient rotation (Figure 2) and different anatomical areas visualized. The final prospective dataset contained 4752 studies from 113 diagnostic devices (Table 1).

The radiologists’ reports of 4752 cases were classified as containing or not containing a description of CXR abnormalities via a home-built machine-based text analysis tool [24]. All cases were analyzed in terms of the target pathologies. Therefore, only a single CXR reading result was used as ground truth. We performed additional testing to confirm the possibility of using a single radiologist reading as the ground truth for the prospective evaluation. A subset of the AI cases from prospective study (378 from 4752) was interpreted by three experts (level of experience in thoracic radiology > 5 years): 97 random cases for which AI results and radiology reports matched and 281 random cases when AI and radiologist’ conclusions deviated (Table 1 and Figure 1). The AI results were classified as normal/abnormal according to the threshold determined on the prospective dataset through ROC analysis. At least two experts interpreted each case with blinding. Cases were assigned to the experts randomly. In case of disagreement between the two experts, a third expert made the final decision, being aware of the opinions of the two experts.

### 2.4. Statistical Analysis

The performance of the AI system was assessed by generating a receiver operating characteristic (ROC) curve from the AI system and radiologist scores. The area under the ROC curve is reported with 95% confidence intervals (CI 95%). Similarly, reader performance was evaluated by thresholding at different score levels to generate ROC points. Confidence intervals were calculated by the Delong method [25]. For ROC analysis, the web tool https://roc-analysis.mosmed.ai/ (accessed on 1 February 2023) was used. The ground true (GT) value was ‘0’ for ‘no pathology’ and ‘1’ for target pathology. To conduct the ROC analysis, the value of the probability of pathology was used as a response from the AI system. For radiologists within the framework of the multi-reader study—also a probabilistic scale. When evaluating the diagnostic accuracy metrics of radiologists from a prospective subsample, binary values of the presence or absence of pathology were used.

The *p*-value was calculated by McNemar’s test. McNemar’s one degree of freedom chi-square test for the equality of proportions is applied for the analysis of matched, binary outcome data [26]. We selected McNemar’s test because AI and human readers analyzed the same data. Finally, the maximum of the Youden Index was used to determine the threshold value for the scores of radiologists and AI [27,28]. For this threshold, sensitivity and specificity were calculated.

The consistency of the binary estimates was determined by an agreement (the proportion of matched estimates) and reliability with Cohen’s kappa coefficient with a CI of 95%. The consistency was calculated between AI, radiologists, and experts.

## 3. Results

### 3.1. Retrospective Evaluation 

The ROC results for all readers and the AI system are shown in Figure 3, green.

The AI system achieved an area under the ROC curve of 0.94 (0.87–1.0)—Figure 3, purple. In most regions of the ROC curve, the system performed a little worse than or at the same level as the average human reader. However, the McNemar test showed no statistically significant differences (Table 2).

### 3.2. Prospective Evaluation

The AUC was 0.84 (0.82–0.86) according to the 4752 studies available for analysis. Thus, the prospective ROC curve recalculated using the Youden index threshold, had a sensitivity of 0.77 (0.73–0.80) and a specificity of 0.81 (0.80–0.82). The ROC curve for determining the presence of pathology is shown in Figure 3, blue.

The prospective evaluation of the AI was based on the study reports with a single reading. To verify the performance of AI in such conditions during the prospective evaluation, a randomized expert review was initiated. The review showed high reliability between experts and radiologists with a Cohen’s kappa equal to 0.79 (0.72–0.85). A Cohen’s kappa ≥ 0.8 corresponds to the “perfect” strength of reliability, according to Landis and Koch [29]. Moreover, during the prospective evaluation, the sensitivity and specificity of radiologists were 0.86 (0.82–0.91) and 0.92 (0.88–0.96), respectively, and matched the averaged multi-reader study radiologist performance within the confidence intervals (Table 3). The three-point ROC results for radiologists on this subset are shown in Figure 3, yellow.

We observed that in expert or consensus-confirmed false-positive results (153 cases), the AI system had detected atelectasis, cardiomegaly, consolidation, fibrosis, and calcifications in the vast majority of cases (139 cases–91%). However, these findings were not reported, either undetected because of small size or unreported due to subjective low clinical significance.

For 104 false-negative cases (81%), reports described either “opacity” (lung inflammation term encompassing consolidation) or “nodule/mass”, as well as calcification.

Selected false-positive (Figure 4a,b), false-negative (Figure 4c,d), and true-positive (Figure 4e), true-negative (Figure 4f) examples are provided in Figure 4.

## 4. Discussion

We determined the diagnostic accuracy of the commercial AI solution (Lunit INSIGHT CXR) for both retrospective (multi-reader multi-case) and prospective studies. This corresponds to the capability (local test set) and durability (prospective study) stages proposed by Larson et al. in their study [30]. It should be noted that the multi-reader study was based on only a limited set of cases, selected explicitly knowing the capabilities and requirements of the AI system and subjected to peer review. Thus, the dataset contained only “benchmark cases” without ambiguous results or technical defects that might be present in routine practice. The bulk of scientific publications on the performance of AI solutions is based on benchmarked, thoroughly verified datasets. We were able to assess the performance of AI in “real” conditions, identical to those in which radiologists work. Our findings suggest that the tested AI system should only be implemented in scenarios of significant staff shortage, as its diagnostic performance had not fully matched the human interpretation in the prospective study. Below, we offer several ways to improve the quality of CXR AI systems.

### 4.1. Additional Data and Technical Assessment of the Quality of Input CXR

Usually, CXR is performed in two projections-frontal and lateral. Thus, a radiologist has an additional source of diagnostic information in ambiguous cases. However, when analyzing the AI results, we excluded cases in which the lateral projection was processed because the AI is only intended for frontal CXR and detects a limited pathology set (10 findings). Therefore, some pathological changes are undetected, including fractures, hiatal hernia, interstitial lung disease, adenopathy, emphysema, pleural plaques, free air under diaphragm, tube or catheter malposition, and foreign bodies. Our results may have been affected by the COVID-19 pandemic, as diagnosing viral pneumonia with pulmonary infiltrates was not available via the evaluated AI system. We believe this flaw has already been fixed in the current version of AI systems. For example, qXR v2.1 c2 (Qure.ai Technologies) is already able to process lateral projection, increasing the accuracy of identifying individual pathological signs [15].

The example with incorrect processing of the lateral projection clearly showed the need for preprocessing the image in order to filter the input data. In the current AI implementation, the system is tag-oriented; if the tags are correct, then the input image is chosen correctly. However, if they contain errors due to the human factor, then the algorithm can be fed an image with a completely different projection or anatomical region. With the mass introduction of autonomous AI systems, it is imperative to implement a quality control system for the input image. We recommend performing quality control of the input data on the AI side to ensure AI safety as a stand-alone regardless of the use case.

Over the past three years, many studies have shown the possibilities for AI application in radiology as CDSS or even as a second reading. Nevertheless, to justify the possibility of using AI algorithms on par with a radiologist, it is necessary to solve several problems [31]. These problems include the lack of a unified methodological framework (external datasets, difficulty preparing datasets and comparing with human performance), lack of standardized nomenclature, and heterogeneous outcome measures (area under the receiver operating characteristic, sensitivity, positive predictive value, and F1 score). We have considered all the requirements from the “Reporting guidelines for clinical trial reports for interventions involving artificial intelligence” [32]. Apart from the AI performance metrics, we focused on error analysis, described how poor or unavailable input data were evaluated and processed, and clarified whether human–AI interaction occurred during input data processing and what skill level was required.

### 4.2. Comprehensive Assessment of Identified Radiographic Features

The high performance of the AI system evaluated in our study has been previously reported [33,34,35]. In our study, we obtained reliable data on sensitivity, specificity, and accuracy. We observed variability in the assessment of the same study by different radiologists. Sometimes mild linear atelectasis, cardiomegaly, and fibrosis were not reported in the “conclusion” part of the report, likely due to their low prognostic value. However, the clinical importance of such findings may be addressed via the integration of clinical data, such as patient gender, age, and frequency of pathological findings in these subgroups, into computer vision algorithms.

For instance, for age-related patients with clinically insignificant changes, such as fibrosis, a radiologist can often write the ‘age norm’ in the conclusion. In fact, the term ‘Age Conditional Norm’ indicates the absence of changes that require additional appointments from the general practitioner. Any AI system currently used in CXR analysis solves two sequential tasks: change detection and its classification. To correctly classify CXR with age-related changes, information about the patient’s age can be fed to the input of the classifier.

In this regard, we would like to give an example of a study by A. Akselrod-Ballin et al. (2019), who, using the example of AI for mammography, showed that incorporating additional clinical data can improve the diagnostic quality of the AI system [36]. An algorithm that can assess the pathology not only on a binary scale (present/absent) but also provides information on its prevalence in the population may have increased clinical value for CXR.

### 4.3. Fine-Tuning of AI System

Although AI can be optimized for a specific task, there are little data in the literature on how to tune them. However, tuning an AI system can be achieved relatively easily using ROC analysis. The output of the AI algorithm is a probability of pathology, and finding the optimal threshold value involves determining the probability value above which the study will be classified as pathological and below as normal. The higher the threshold, the more specific the AI system will be, and the lower, the more sensitive. The question remains, what values of diagnostic accuracy metrics should be considered optimal? Certain guidelines state that a threshold of 0.81 must be exceeded for all diagnostic accuracy metrics [37]. In our multi-reader study on a retrospective set of studies, we obtained significantly higher values: sensitivity of 0.9, specificity of 0.95, and AUC of 0.97. According to these AI metrics, the system did not differ significantly from the average radiologist. This indicates the possibility of setting a minimum bar for diagnostic input accuracy for AI systems on a limited dataset of 0.9. The latest review [38] shows that of the 46 AI systems examined, 18 (39%) exceed 0.9 in terms of diagnostic accuracy metrics.

Evaluation of the prospective study showed a decrease in the diagnostic accuracy metrics of the AI system compared to a radiologist. In this case, the following solution is possible: changing the threshold of the AI to match either the sensitivity of the radiologist or its specificity. When maximizing the sensitivity of the AI algorithm, we believe that it is possible to use it for primary CXR reading to sort out studies without pathology. The increase in negative predictive value would lead to patient list optimization and, possibly, reduce the professional burnout of radiologists. The proposed use case requires a separate study with simulated triage, analogous to Sverzellati N. et al. (2021) [39].

Considering practical options for AI application, several conclusions can be drawn. First, AI-based first reading can be a viable option in cases of staff shortages, provided that the AI is calibrated to achieve a sensitivity level comparable to that of radiologists. However, further improvements are needed to increase the diagnostic value of the AI system in clinical practice. Additionally, prioritizing AI findings based on their clinical significance, particularly in cases where multiple findings are present, is recommended. For instance, the presence of a single calcification or fibrosis may not have significant clinical value, whereas the presence of calcification and fibrosis, particularly in the upper and middle lung lobes, could indicate pulmonary tuberculosis, a socially significant disease that is routinely screened for in different countries.

## 5. Conclusions

In a large-scale prospective validation of the commercial AI algorithm in clinical practice, lower sensitivity and specificity values were obtained compared to a retrospective study on the local test set. The lower prospective performance was associated with an increased number of findings considered by experts to be clinically insignificant, including mild atelectasis, cardiomegaly, and consolidation.

## Figures and Tables

**Figure 1 diagnostics-13-01430-f001:**
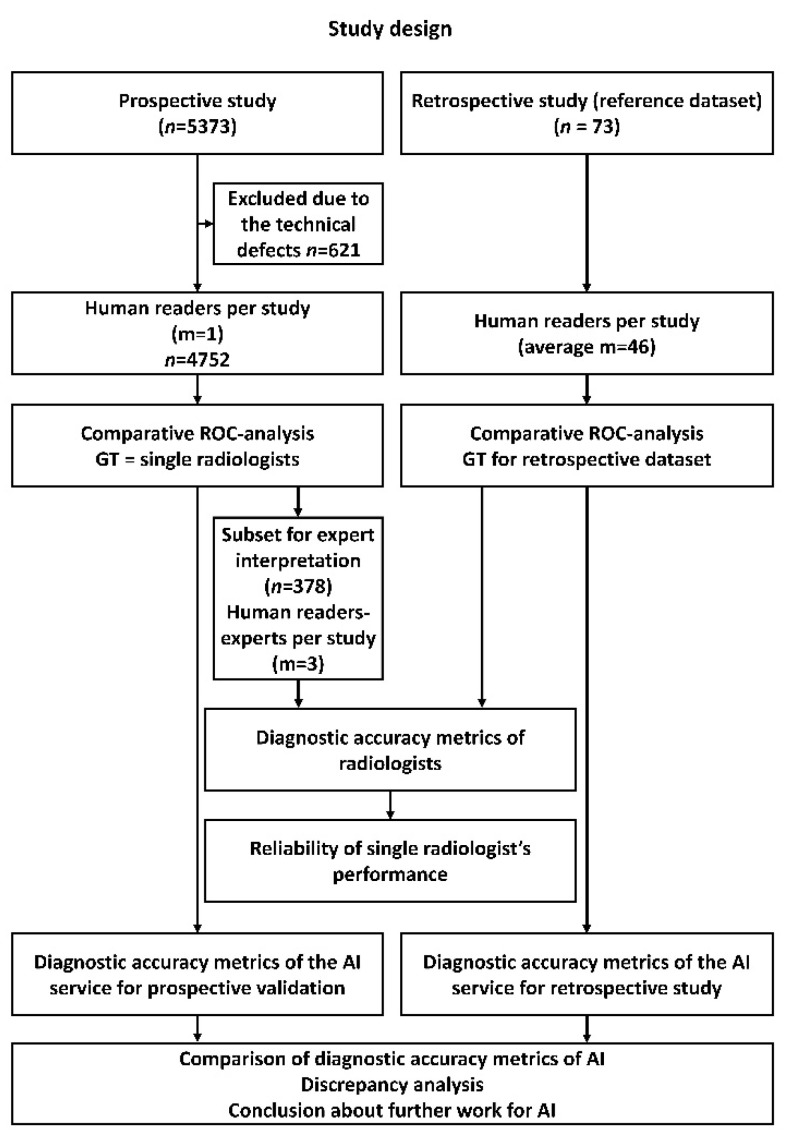
The study design flowchart. GT-ground truth; m—amount of human readers; *n*—amount of studies.

**Figure 2 diagnostics-13-01430-f002:**
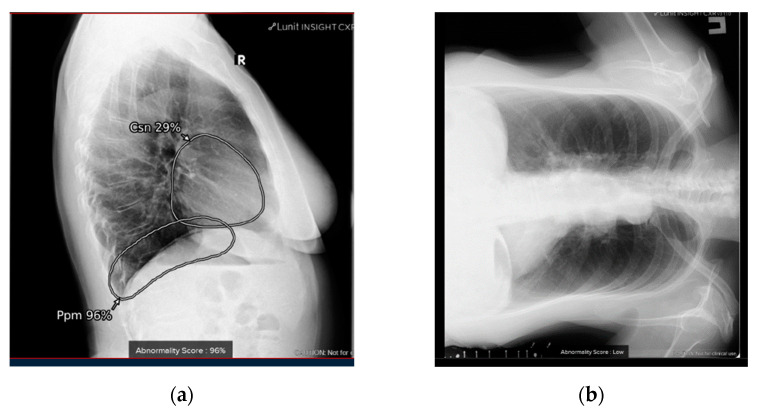
Examples of the technical defects: lateral instead of frontal projection was analyzed by the AI system (**a**); the image with relevant radiological findings is rotated, and the study was reported by a radiologist as having right lower lobe opacity (**b**).

**Figure 3 diagnostics-13-01430-f003:**
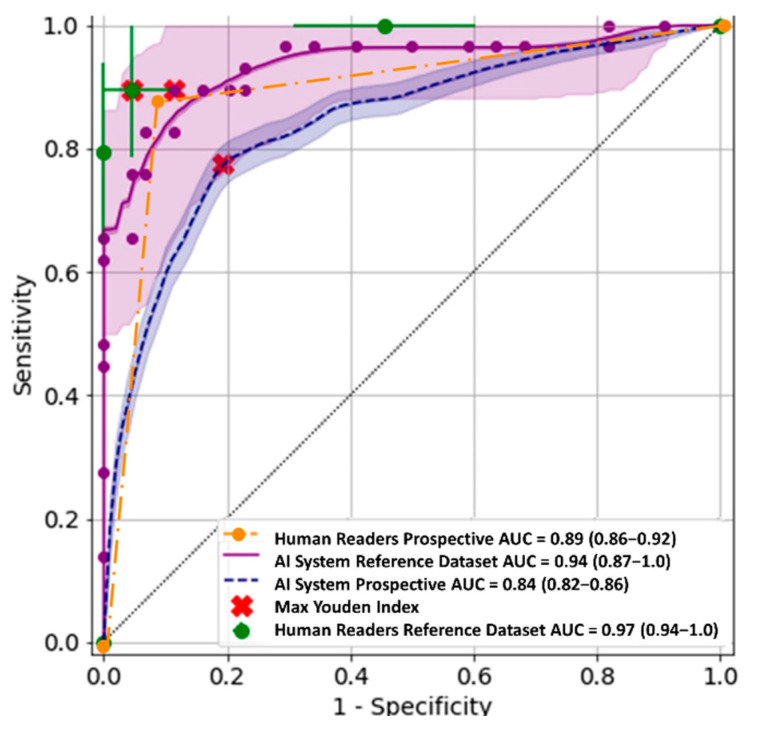
Receiver operator characteristic curves for multi-reader study (green) and AI performance on the local test set (purple), prospective performance of AI (blue). Receiver operator characteristic curve for radiologists on the sample from the prospective dataset for expert interpretation (yellow).

**Figure 4 diagnostics-13-01430-f004:**
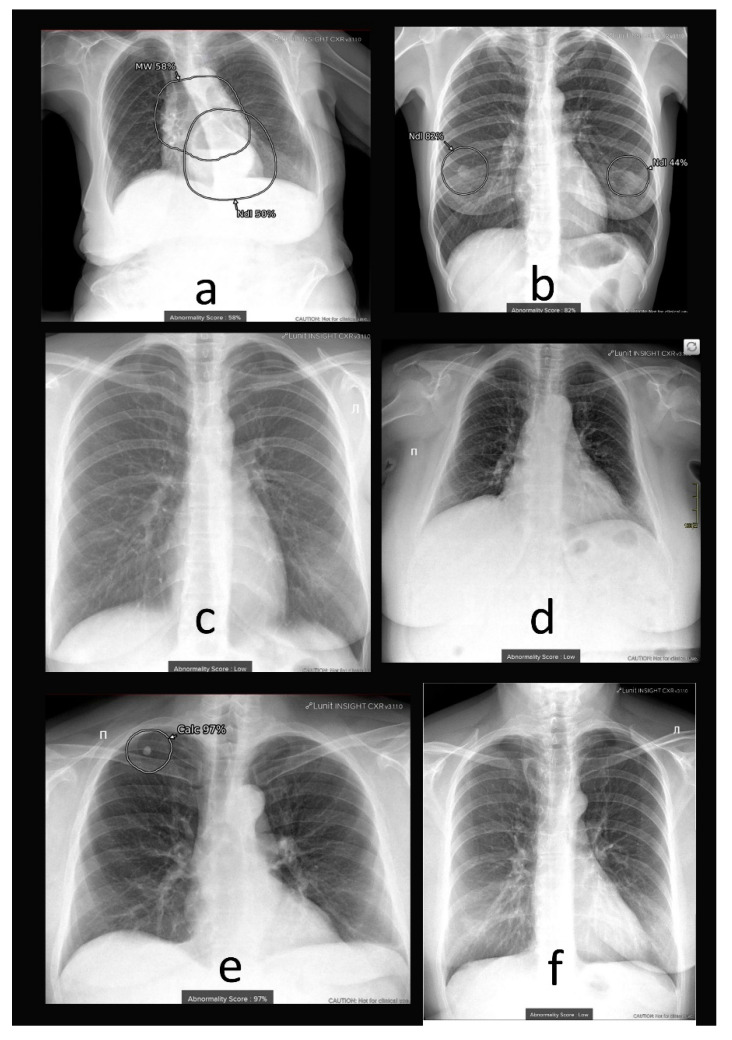
Examples of the (**a**,**b**) false-positive and (**c**,**d**) false-negative and (**e**) true-positive, (**f**) true-negative cases of the AI system: (**a**) patient rotation and hiatal hernia, AI erroneously detected mediastinal widening and nodule; (**b**) female patient with visible nipples, AI detected right-sided lung nodule with high probability and left-sided nodule with lower probability, but radiologist did not report any pathology; (**c**) left-sided opacity reported by a radiologist and not reported by AI; (**d**) cardiomegaly reported by radiologist and not reported by AI; (**e**) a nodule at the apex of the right lung (calcinate); (**f**) study without pathology, correctly reported by AI, but erroneously reported by radiologist as an inflammatory process in the right lung (experts did not report pathological changes, which was confirmed by subsequent computed tomography).

**Table 1 diagnostics-13-01430-t001:** Details on the retrospective local test set and prospective dataset.

Characteristic	Local Test Set	Prospective Dataset	Sample from the Prospective Dataset Additionally Interpreted by Experts
No. of patients (abnormal * cases)	73 (29)	4752 (694)	378 (185)
Number of radiologists	160 (average 46 per study)	226 (1 per study)	99 (1 per study)
Confirmation of (ab)normality by	consensus two experts (>5 years of experience)	radiology report	consensus of at least two experts (>5 years of experience)
Male/female/unknown	30/42/1	1746/3005/1	140/238
Age (y) **	50 ± 19	49 ± 16	55 ± 17
No. of diagnostic devices	38	113	79
Vendors	GE HealthCare, Chicago, IL, USA S.P.Helpic LLC, Moscow, Russia Fujifilm, Tokyo, Japan Philips, Amsterdam, The Netherlands Toshiba Medical Systems Corporation, Tokyo, Japan NIPK Electron Co., Saint Petersburg, Russia MEDICAL TECHNOLOGIES Co., Ltd., Moscow, Russia.		

* Abnormal case contained at least one of the following radiologic findings: (1) pleural effusion; (2) pneumothorax; (3) atelectasis; (4) opacity; (5) opacity/consolidation; (6) dissemination (>20 focal abnormalities); (7) cavity with air; (8) cavity with fluid level; (9) pulmonary calcification; (10) bone fracture. ** Data are mean + standard deviation. Data in parentheses are the range.

**Table 2 diagnostics-13-01430-t002:** The human and AI system diagnostic performance metrics: multi-reader study.

Diagnostic Performance Metrics	AI System	Human Readers	*p*-Value
AUC (CI 95%)	0.94 (0.87–1.0)	0.97 (0.94–1.0)	0.51
Sensitivity * (CI 95%)	0.9 (0.79–1.0)	0.9 (0.79–1.0)	1.0
Specificity * (CI 95%)	0.89 (0.79–0.98)	0.95 (0.89–1.0)	0.26
Cappa Kohen * for GT	0.74 (0.58–0.9)	0.86 (0.74–0.98)	
Agreement * for GT	93%	88%	
Cohen’s Kappa * for radiologists and AI	0.71 (0.54–0.88)		
Agreement * for radiologists and AI	86%		

* at the operating point of the maximum Youden index. The threshold for AI was 0.5 (Youden index was 0.78), and 3 for human readers (Youden index was 0.85).

**Table 3 diagnostics-13-01430-t003:** The human and AI system diagnostic performance metrics: prospective dataset.

Diagnostic Performance Metrics	AI System	Human Readers (Sample from the Prospective Dataset for Expert Interpretation)
AUC (CI 95%)	0.84 (0.82–0.86)	0.89 (0.86–0.92)
Sensitivity (CI 95%)	0.77 * (0.73–0.80)	0.86 ** (0.82–0.91)
Specificity (CI 95%)	0.81 * (0.80–0.82)	0.92 ** (0.88–0.96)
Cappa Kohen * for radiologists and AI	0.42 (0.38–0.45)	
Agreement * for radiologists and AI	81%	
Cohen’s Kappa ** for radiologists and experts		0.79 (0.72–0.85)
Agreement ** for radiologists and experts		89%

* at the operating point of the maximum Youden index. The threshold for AI was 0.17 (Youden index was 0.58). ** at the operating point of the maximum Youden index. The threshold for human readers was 1 (Youden index was 0.79).

## Data Availability

The data presented in this study are available upon request from here: [https://mosmed.ai/en/datasets/ (accessed on 4 April 2023)] or from the corresponding author. The datasets are not publicly available due to the fact that the datasets, on which the study was conducted, are still used to test AI systems participating in the experiment on the use of innovative computer vision technologies for medical image analysis and subsequent applicability in the healthcare system of Moscow.

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
