# Peer review of "AI-Based CXR First Reading: Current Limitations to Ensure Practical Value"

_diagnostics, 2023, doi:10.3390/diagnostics13081430_

Round 1

Reviewer 1 Report

The paper is well-written and organized. The experiments with radiologists and the details are given. I have one question. What is the AI system in the paper? Could you give more details?

Please revise the Scheme I. The authors can draw it more readable with Microsoft Visual Studio or any other programs.

Author Response

Point 1: The paper is well-written and organized. The experiments with radiologists and the details are given. I have one question. What is the AI system in the paper? Could you give more details?.

Response 1: We thank the reviewer for the high evaluation of our paper.

In this work, we used the commercial AI system Lunit INSIGHT CXR. This system used a ResNet34-based deep convolutional neural network with 10 different abnor-mality-specific channels in the final layer [DOI: 10.1183/13993003.03061-2020]. Detailed information about the AI system is available on the official web-site [https://www.lunit.io/en/products/cxr].

This information has been added to the Materials and Methods section.

Point 2: Please revise the Scheme I. The authors can draw it more readable with Microsoft Visual Studio or any other programs.

Response 2: We thank the reviewer for the suggestion.

Scheme I has been revised for the convenience of the reader and saved as vector graphics. An export image with a resolution of 600 dpi, as well as an svg file will be attached to the article.

Reviewer 2 Report

AI-based CXR first reading: current limitations to ensure practical value

In this article, performed a multicenter external evaluation of the practical and clinical efficacy of a commercial AI algorithm for chest X-ray (CXR) analysis (Lunit INSIGHT CXR). A retrospective evaluation was performed with a multi-reader multi-case (MRMC) study. For a prospective evaluation, the AI model was run on CXR studies; the results were compared to the reports of 226 radiologists.

To me this short report on AI-based CXR first reading is enough for the publication in this journal. Few points are necessary to improve this work, given as:

1.      Statistics given in the table 1, lacks of references. You should add the proper reference, from where you found this data.

2.      Figure 1 is blur, it is difficult for a reader to read and understand the content.

3.      Results and Conclusion section is very short. Add more content to these section to improve this work.

4.      Reference style is very haphazard, kindly make it in a single format or use the journal designed bibliography style.

5.      The literature about diseases is very limited, I recommend to add more detail about topic. I recommend to add the following, Disease categorization with clinical data using optimized bat algorithm and fuzzy value, Breast Cancer Chemical Structures and their Partition Resolvability, Vertex Metric Resolvability of COVID Antiviral Drug Structures, Partition Dimension of COVID Antiviral Drug Structures.

Author Response

We thank the reviewer for dedicating time to review our work.

Your detailed and insightful comments have been particularly helpful in identifying areas where our manuscript could be strengthened, and we have carefully considered all of your suggestions. We believe that your feedback has significantly improved the manuscript and we are grateful for your efforts.

We agree with the reviewer that the first part of the article (MRMC) is of great interest in itself. However, in practise, during the experiment [https://mosmed.ai/en/ai/], we found that AI systems, when working in a stream, do not achieve the diagnostic accuracy metrics obtained during validation on a limited data set. Therefore, we consider it valuable to publish these results in one article. In our opinion, such an article will be more practice-oriented and will improve the quality of AI systems for medical diagnostics.

Point 1: Statistics given in the table 1, lacks of references. You should add the proper reference, from where you found this data.

Response 1: We apologize for the possible confusion. We have added the references for positions in table 1.

Point 2: Figure 1 is blur, it is difficult for a reader to read and understand the content.

Response 2: We apologize for difficulties to observe the Figure 1 and have modified it.. An export image with a resolution of 600 dpi, as well as an svg file will be attached to the article.

Figure 1. The study design flowchart. GT-ground truth; m – amount of human readers; n – amount of studies.

Point 3: Results and Conclusion section is very short. Add more content to these section to improve this work.

Response 3: Thank you for your comment. We have modified the discussion section of the manuscript for a clearer understanding of the results and conclusions.

We determined the diagnostic accuracy of the commercial AI solution (Lunit IN-SIGHT CXR) for both retrospective (multi-reader multi-case) and prospective studies. This corresponds to the capability (local test set) and durability (prospective study) stages proposed by Larson et al. in their study [30]. It should be noted that the mul-ti-reader study  was based on only a limited set of cases, selected explicitly knowing the capabilities and requirements for AI system and subjected to peer review. Thus, the dataset contained only "benchmark cases" without ambiguous results or technical de-fects that might be present in routine practice. The bulk of scientific publications on the performance of AI solutions are based on benchmarked, thoroughly verified datasets. We were able to assess the performance of AI in "real" conditions, identical to those in which radiologists work. Our findings suggest that the tested AI system should only be implemented in scenarios of significant staff shortage, as its diagnostic performance had not fully matched the human interpretation in the prospective study.

Usually, CXR is performed in two projections - frontal and lateral. Thus, a radiol-ogist has an additional source of diagnostic information in ambiguous cases. However, when analyzing the AI results, we excluded cases in which the lateral projection was processed because the AI is only intended for frontal CXR and detects a limited pa-thology set (10 findings). Therefore, some pathological changes are undetected, in-cluding fractures, hiatal hernia, interstitial lung disease, adenopathy, emphysema, pleural plaques, free air under diaphragm, tube or catheter malposition, foreign bodies. Our results may have been affected by the COVID-19 pandemic, as diagnosing viral pneumonia with pulmonary infiltrates were not available via the evaluated AI system. We believe this flaw has already been fixed in the current version of AI systems. For example, qXR v2.1 c2 (Qure.ai Technologies) is already able to process the lateral pro-jection, increasing the accuracy of identifying individual pathological signs [15].

The example with incorrect processing of the lateral projection clearly showed the need for preprocessing the image in order to filter the input data. In the current AI im-plementation, the system is tag-oriented: if the tags are correct, then the input image is chosen correctly. But if they contain errors due to the human factor, then the algorithm can be fed an image with a completely different projection or anatomical region. With the mass introduction of autonomous AI systems, it is imperative to implement a qual-ity control system for the input image. We recommend performing quality control of the input data on the AI side to ensure AI safety as a stand-alone regardless of the use case.

Over the past three years, many studies have shown the possibilities for AI appli-cation in radiology as CDSS or even as a second reading. Nevertheless, to justify the possibility of using AI algorithms on par with a radiologist, it is necessary to solve sev-eral problems [31]. These problems include the lack of a unified methodological framework (external datasets, difficulty preparing datasets and comparing with hu-man performance), lack of standardized nomenclature, and heterogeneous outcome measures (area under receiver operating characteristic, sensitivity, positive predictive value, and F1 score). We have considered all the requirements from the "Reporting guidelines for clinical trial reports for interventions involving artificial intelligence" [32]. Apart from the AI performance metrics, we focused on error analysis, described how poor or unavailable input data were evaluated and processed, clarified whether human-AI interaction occurred during input data processing and what skill level was required.

The high performance of the AI system evaluated in our study has been previously reported [33-35]. In our study, we obtained reliable data on sensitivity, specificity, and accuracy. We observed variability in the assessment of the same study by different ra-diologists. Sometimes mild linear atelectasis, cardiomegaly, and fibrosis were not re-ported in the "conclusion" part of the report, likely due to their low prognostic value. However, the clinical importance of such findings may be addressed via the integra-tion of clinical data, such as patient gender, age, and frequency of pathological find-ings in these subgroups, into computer vision algorithms.

For instance, for age-related patients with clinically insignificant changes, such as fibrosis, a radiologist can often write the ‘age norm’ in the conclusion. In fact, the term ‘Age Conditional Norm’ indicates the absence of changes that require additional ap-pointments from the general practitioner. Any AI system currently used in CXR analy-sis solves two sequential tasks: change detection and its classification. To correctly classify CXR with age-related changes, information about the patient's age can be fed to the input of the classifier.

In this regard, we would like to give an example of a study by A. Akselrod-Ballin et al. (2019), who, using the example of AI for mammography, showed that incorpo-rating additional clinical data can improve the diagnostic quality of the AI system [36]. An algorithm that can assess the pathology not only on a binary scale (pre-sent/absent) but also provides information on its prevalence in the population may have increased clinical value for CXR..

 Although AI can be optimized for a specific task, there is little data in the litera-ture on how to tune them. However, tuning an AI system can be achieved relatively easily using ROC analysis. The output of the AI algorithm is a probability of pathology, and finding the optimal threshold value involves determining the probability value above which the study will be classified as pathological and below as normal. The higher the threshold, the more specific the AI system will be, and the lower, the more sensitive. The question remains, what values of diagnostic accuracy metrics should be considered optimal? Certain guidelines state that a threshold of 0.81 must be exceeded for all diagnostic accuracy metrics [37]. In our multi-reader study on a retrospective set of studies, we obtained significantly higher values: sensitivity of 0.9, specificity of 0.95, and AUC of 0.97. According to these AI metrics, the system did not differ signifi-cantly from the average radiologist. This indicates the possibility of setting a minimum bar for diagnostic input accuracy for AI systems on a limited data set of 0.9. The latest review [38] shows that of the 46 AI systems examined, 18 (39%) exceed 0.9 in terms of diagnostic accuracy metrics.

Evaluation of the prospective study showed a decrease in the diagnostic accuracy metrics of the AI system compared to a radiologist. In this case, the following solution is possible: changing the threshold of the AI to match either the sensitivity of the radi-ologist or its specificity. When maximizing the sensitivity of the AI algorithm, we be-lieve that it is possible to use it for primary CXR reading to sort out studies without pathology. The increase in negative predictive value would lead to patient list optimi-zation and, possibly, reduce the professional burnout of radiologists. The proposed use case requires a separate study with simulated triage, analogous to Sverzellati N. et al. (2021) [39].

Considering practical options for AI application, several conclusions can be drawn. First, AI-based first reading can be a viable option in cases of staff shortages, provided that the AI is calibrated to achieve a sensitivity level comparable to that of radiologists. However, further improvements are needed to increase the diagnostic value of the AI system in clinical practice. Additionally, prioritizing AI findings based on their clinical significance, particularly in cases where multiple findings are present, is recommended. For instance, the presence of a single calcification or fibrosis may not have significant clinical value, whereas the presence of calcification and fibrosis, par-ticularly in the upper and middle lung lobes, could indicate pulmonary tuberculosis, a socially significant disease that is routinely screened for in different countries.

Point 4: Reference style is very haphazard, kindly make it in a single format or use the journal designed bibliography style.

Response 4: Thank you for bringing this to our attention. We have corrected the sections with the bibliography.

Point 5: The literature about diseases is very limited, I recommend to add more detail about topic. I recommend to add the following, Disease categorization with clinical data using optimized bat algorithm and fuzzy value, Breast Cancer Chemical Structures and their Partition Resolvability, Vertex Metric Resolvability of COVID Antiviral Drug Structures, Partition Dimension of COVID Antiviral Drug Structures..

Response 5: Thank you for your comment. The purpose of this work was not to evaluate the clinical picture of the disease. We have used only radiographic patterns that radiologists use when describing a study. Throughout the study, we used a binary assessment: if the study had a pathological feature identified by this AI system, then this study was assigned a class "1" (pathology), otherwise "0" (normal). We did not evaluate the accuracy of identifying individual radiological patterns.

Reviewer 3 Report

Manuscript Title “AI-based CXR first reading: current limitations to ensure practical value”

General comment:

The quality of this manuscript is OK, however, the structure of this manuscript has some inappropriate arrangement, thus, it needs to largely revise and emphasize the technical term of the calculated algorithm. The specific comments are listed below

Specific comment:

1.      Abstract; OK, however, the last sentence is giving too aggressive to imply the result

2.      Introduction; too rough to illustrate the background review and rationale study

3.      Materials and methods; too weak the technical description to imply the MRMC algorithm, plus, there is no solid description of the flowchart as depicted in scheme 1 (suggested to redefine as figure 1 to simplify the naming)

4.      How do you define the accuracy, sensitivity or specificity in the algorithm as listed in Table 2

5.      Results; why figure 2a and 2b cannot be joined together to simplify the illustration

6.       Discussion; strongly suggested to categorize the content into several sub-sections with specific title.

7.      L250-259, the drawbacks of adopting this algorithm should be elaborated more to offer the solid idea in practical application

8.      L260, “Considering the options for practical AI application, the following conclusions can be made: AI-based first reading in staff shortage is possible, provided the AI is tuned to a target sensitivity comparable to that of the radiologist.” sounds too aggressive as well.

9.      L262, “It is necessary to further improve the AI service to provide diagnostic value in clinical practice. We also consider it advisable to refine the AI finding prioritization according to their clinical significance, especially if multiple findings are present.” This one look OK though

10.  Conclusion is OK

Author Response

Point 1: Abstract; OK, however, the last sentence is giving too aggressive to imply the result.

Response 1: Thank you for your remark. With this proposal, we wanted to show that in case of a shortage of radiologists, we can rekommend to use this version of the AI system. However, according to information from the developer, a new version of the algorithm is already available, so we decided to remove the last sentence.

Point 2: Introduction; too rough to illustrate the background review and rationale study.

Response 2: Thank you for your comment. We have updated the introduction by adding the following information:

The diagnostic accuracy of the algorithms provided by the developers is quite high [7–9], reaching the same accuracy for radiologists [10], and for some solutions even exceeding them [11, 12]. As of the beginning of 2023, 29 AI-based software prod-ucts have European certification for medical use as a medical device (CE MDR/MDD), of which 11 have passed similar certification in the United States [13]. It is important to note that one such software product, approved for circulation as a medical device, is intended for a completely autonomous analysis of CXR [14]. This AI algorithm sorts examinations, detects CXR without pathology, forms a complete description protocol that does not require validation by a radiologist; this approach reduces the burden on the radiologist, allowing to focus on cases with pathologies [11].

The problem is that most of the work on assessing the diagnostic accuracy of AI algorithms for CXR indicates metrics obtained by developers on limited data sets in the so-called "laboratory conditions". As can be seen from recent studies [12, 15], the met-rics obtained in this way look attractive for the subsequent implementation of such algorithms in clinical practice. Will AI for CXR analysis also work well and demon-strate high diagnostic accuracy metrics in real clinical practice? There are very few large-scale works on this topic; it is possible to point out only a few studies [11, 16]. Using recent research as an example, we can see the introduction of autonomous AI into clinical practice, but the issue of threshold values for diagnostic accuracy metrics remains open.

Point 3: Materials and methods; too weak the technical description to imply the MRMC algorithm, plus, there is no solid description of the flowchart as depicted in scheme 1 (suggested to redefine as figure 1 to simplify the naming).

Response 3: Thank you for your remark. We used simplified version of MRMC. It was an international multi-reader study, thar has been performed to determine an average diagnostic accuracy of radiologists interpreting chest x-ray images for benchmarking with the stand-alone AI system performance metrics for the same use cases. Olso we have added the additional imformation, describing the design of study. Scheme I was redefined as Figure 1.

Point 4:  How do you define the accuracy, sensitivity or specificity in the algorithm as listed in Table 2.

Response 4: Thank you for your comment. We have modified the Methods section accordingly.

The performance of the AI system was assessed by generating a receiver operating characteristic (ROC) curve from the AI system and radiologist scores. The area under the ROC curve is reported with 95% confidence intervals (CI 95%). Similarly, reader performance was evaluated by thresholding at different score levels to generate ROC points. Confidence intervals were calculated by the Delong method [25]. For ROC analysis, web tool https://roc-analysis.mosmed.ai/ was used. The ground true (GT) val-ue was ‘0’ for ‘no pathology’ and ‘1’ for target pathology. To conduct the ROC analysis, the value of the probability of pathology was used as a response from the AI system. For radiologists within the framework of the multi-reader study- also a probabilistic scale. When evaluating the diagnostic accuracy metrics of radiologists from a prospec-tive subsample, binary values of the presence or absence of pathology were used.

Maximum of Youden Index was used to determine the threshold value for the scores of radiologists and AI [27, 28]. For this threshold, sensitivity and specificity were calculated. 

Point 5: Results; why figure 2a and 2b cannot be joined together to simplify the illustration.

Response 5: We deliberately separated these two graphs to be able to compare the diagnostic accuracy metrics of the AI and the radiologist. Figure 2B shows the ROC curve for a subsample for radiologists control, for which it is incorrect to calculate the results of the AI system, therefore, in order not to confuse the reader, we divided it into two separate images.

Point 6: Discussion; strongly suggested to categorize the content into several sub-sections with specific title..

Response 6: We followed your advice and placed all the ROC curves on one chart.

Point 7: L250-259, the drawbacks of adopting this algorithm should be elaborated more to offer the solid idea in practical application.

Response 7: Thank you for your comment. We have modified this part of Discussion section accordingly.

The high performance of the AI system evaluated in our study has been previously reported [33-35]. In our study, we obtained reliable data on sensitivity, specificity, and accuracy. We observed variability in the assessment of the same study by different radiologists. Sometimes mild linear atelectasis, cardiomegaly, and fibrosis were not reported in the "conclusion" part of the report, likely due to their low prognostic value. However, the clinical importance of such findings may be addressed via the integration of clinical data, such as patient gender, age, and frequency of pathological findings in these subgroups, into computer vision algorithms.

For instance, for age-related patients with clinically insignificant changes, such as fibrosis, a radiologist can often write the ‘age norm’ in the conclusion. In fact, the term ‘Age Conditional Norm’ indicates the absence of changes that require additional appointments from the general practitioner. Any AI system currently used in CXR analysis solves two sequential tasks: change detection and its classification. To correctly classify CXR with age-related changes, information about the patient's age can be fed to the input of the classifier.

In this regard, we would like to give an example of a study by A. Akselrod-Ballin et al. (2019), who, using the example of AI for mammography, showed that incorporating additional clinical data can improve the diagnostic quality of the AI system [36]. An algorithm that can assess the pathology not only on a binary scale (present/absent) but also provides information on its prevalence in the population may have increased clinical value for CXR.

Point 8: L260, “Considering the options for practical AI application, the following conclusions can be made: AI-based first reading in staff shortage is possible, provided the AI is tuned to a target sensitivity comparable to that of the radiologist.” sounds too aggressive as well..

Response 8: Thank you for your comment. We have modified the part of Discussion section accordingly.

Although AI can be optimized for a specific task, there is little data in the literature on how to tune them. However, tuning an AI system can be achieved relatively easily us-ing ROC analysis. The output of the AI algorithm is a probability of pathology, and finding the optimal threshold value involves determining the probability value above which the study will be classified as pathological and below as normal. The higher the threshold, the more specific the AI system will be, and the lower, the more sensitive. The question remains, what values of diagnostic accuracy metrics should be consid-ered optimal? Certain guidelines state that a threshold of 0.81 must be exceeded for all diagnostic accuracy metrics [37]. In our multi-reader study on a retrospective set of studies, we obtained significantly higher values: sensitivity of 0.9, specificity of 0.95, and AUC of 0.97. According to these AI metrics, the system did not differ significantly from the average radiologist. This indicates the possibility of setting a minimum bar for diagnostic input accuracy for AI systems on a limited data set of 0.9. The latest re-view [38] shows that of the 46 AI systems examined, 18 (39%) exceed 0.9 in terms of diagnostic accuracy metrics.

Evaluation of the prospective study showed a decrease in the diagnostic accuracy metrics of the AI system compared to a radiologist. In this case, the following solution is possible: changing the threshold of the AI to match either the sensitivity of the radi-ologist or its specificity. When maximizing the sensitivity of the AI algorithm, we be-lieve that it is possible to use it for primary CXR reading to sort out studies without pathology. The increase in negative predictive value would lead to patient list optimi-zation and, possibly, reduce the professional burnout of radiologists. The proposed use case requires a separate study with simulated triage, analogous to Sverzellati N. et al. (2021) [39].

Considering practical options for AI application, several conclusions can be drawn. First, AI-based first reading can be a viable option in cases of staff shortages, provided that the AI is calibrated to achieve a sensitivity level comparable to that of radiologists. However, further improvements are needed to increase the diagnostic value of the AI system in clinical practice. Additionally, prioritizing AI findings based on their clinical significance, particularly in cases where multiple findings are present, is recommended. For instance, the presence of a single calcification or fibrosis may not have significant clinical value, whereas the presence of calcification and fibrosis, par-ticularly in the upper and middle lung lobes, could indicate pulmonary tuberculosis, a socially significant disease that is routinely screened for in different countries.

Point 9: L262, “It is necessary to further improve the AI service to provide diagnostic value in clinical practice. We also consider it advisable to refine the AI finding prioritization according to their clinical significance, especially if multiple findings are present.” This one look OK though.

Response 9: Thank you, we have done some changes to the sentence structure to improve the clarity.

However, further improvements are needed to increase the diagnostic value of the AI system in clinical practice. Additionally, prioritizing AI findings based on their clinical significance, particularly in cases where multiple findings are present, is recommended. For instance, the presence of a single calcification or fibrosis may not have significant clinical value, whereas the presence of calcification and fibrosis, par-ticularly in the upper and middle lung lobes, could indicate pulmonary tuberculosis, a socially significant disease that is routinely screened for in different countries.

Point 10: Conclusion is OK.

Response 10: Thank you very much for taking the time to review our manuscript and for providing us with valuable feedback. We greatly appreciate your thoughtful comments and suggestions, which have helped to improve the quality and clarity of our work.

Round 2

Reviewer 1 Report

The paper is now acceptable in the present form. My decision is accept.

Author Response

Thank you for your support of our research. Once again, we are grateful for your time and effort in reviewing our work.

Reviewer 3 Report

The revised manuscript is OK to accept, only one thing that there is still no sub-section head assigned to clarify the topic in the discussion. The others are all OK to hold as in its revised version.

Author Response

Thank you very much for your time and effort in reviewing our manuscript. Your comments and suggestions were incredibly valuable and helped to improve the quality of our research. We greatly appreciate your attention to detail and your insightful feedback.

 Point 1: The revised manuscript is OK to accept, only one thing that there is still no sub-section head assigned to clarify the topic in the discussion. The others are all OK to hold as in its revised version.

 Response 1: We apologize for the possible confusion. We have added the sub-section heads for discussion section:

  • Additional data and technical assessment of the quality of input CXR
  • Comprehensive assessment of identified radiographic features
  • Fine-tuning of AI system

The full text of the discussion section is presented in the attachment.
